# Limits of life: Thermal tolerance of deep-sea hydrothermal vent copepods and implications for community succession

Alessandro Messora[1]*, Stephane Hourdez[2], Monika Bright[3], Teresa Winter[3], Fanny Sieler[1¤], Sabine Gollner[1]

**1** Department of Ocean Systems, NIOZ Royal Netherlands Institute for Sea Research, 't Horntje (Texel), The Netherlands, **2** Laboratoire d'Ecogéochimie des Environnements Benthiques, Observatoire Océanologique de Banyuls, Sorbonne Université-CNRS, Banyuls-sur-Mer, France, **3** Department of Functional and Evolutionary Ecology, University of Vienna, Vienna, Austria

¤ Current address: Leibniz Institute for the Analysis of Biodiversity Change, Hamburg, Germany
* alessandro.messora@nioz.nl

## Abstract

Organisms that live in extreme marine environments naturally experience intermittent exposures to the limits of their physiological potential at different time scales and have developed diverse strategies to survive these variations. We tested the tolerance to thermal stress of deep-sea dirivultid copepod communities from focused and diffuse flows at East Pacific Rise 9°50'N hydrothermal vents in relation to habitat type, oxygen concentration and habitat pressure to unravel their physiological limits to extreme temperature. Lethal median time and temperature experiments were performed to derive the respective thermal death time (TDT) curves. Results showed that dirivultid copepods possess high thermal tolerance exclusively for short exposures and that *in situ* vent fluid flow conditions were an important predictor for maximum tolerated temperatures. Anoxia had a major negative impact on vent copepod survival, whereas atmospheric pressure did not have a significant effect. Results for the upper thermal tolerance of copepods were remarkably similar to macro- and megafauna from the same habitats, while tolerance to hypoxia or anoxia seems to increase with size. Data on relative abundance of dirivultid copepods in their habitats over the past two decades, coupled with data on temperature and anoxia tolerance, suggest that physiological limits strongly impact copepod community composition at focused flow habitats regardless of successional stage. In contrast, complex interplays of interspecific competition, food-source partitioning and experienced small-scale environmental heterogeneity within megafauna aggregations might shape dirivultid community dynamics in diffuse flow habitats.

**Data availability statement:** All relevant data are within the paper and its Supporting Information files.

**Funding:** This study was financially supported by the Dutch Research Council (NWO) (https://www.nwo.nl) through project SUBLIFE in the form of a grant (OCENW.M.22.080; 46520) received by SG. No additional external funding was received for this study. The funder had no role in study design, data collection and analysis, decision to publish, or preparation of the manuscript.

**Competing interests:** The authors have declared that no competing interests exist.

## Introduction

In extreme marine environments, organisms naturally experience intermittent exposures to the limits of their physiological potential at different time scales and have developed diverse strategies to survive these variations [1]. Hydrothermal vents are oasis-like ecosystems located discontinuously along oceanic ridges, volcanic arcs and back-arc basins [2]. Highly specialized organisms aggregate where turbulent mixing provides favourable chemical and thermal conditions [3]. The animal community is supported by chemosynthetic microbes that rely on reduced chemicals emitted from vents for primary production, such as $CH_4$, $H_2$, $Fe^{2+}$ and $H_2S$ [4,5]. Subsequently most larger vent animals and their associated meiofauna have adapted to survive in conditions favourable for chemosynthetic bacteria but inhospitable for most eukaryotic life [6].

Along the East Pacific Rise (EPR), the tubicolous Pompeii worm *Alvinella pompejana* [7,8] thrives in focused flow habitats, where the hot hydrothermal fluids (up to 120°C) at the smoker's surface undergo a substantial temperature decrease to 30°C, helped by the colony's physical and chemical gradients, and mix with the surrounding 2°C seawater [9,10] (Pompeii worm aggregations, Fig 1A). In a large area surrounding black smokers, an extensive subterranean plumbing system provides passage for warm and less acidic diffuse fluids [11] to exit from cracks in the seafloor at temperatures ≤ 35°C [10,12]. Along the EPR these diffuse flows can support aggregations of the giant tubeworm *Riftia pachyptila* and mussel *Bathymodiolus thermophilus* that can thrive at mean temperatures of 2 to 40°C [8,13–15] (tubeworm & mussel aggregations, Fig 1B).

All deep-sea hydrothermal vent habitats are characterized by mixing of vent fluids with cold seawater, resulting in highly variable temperatures that can change within second-long timescales [4]. In addition, vents are subject to a long-term cycle of smothering and (re)colonization following periodic volcanic eruptions [16]. At hydrothermal vents located along 9°50' North EPR, volcanic eruptions frequently occur and were reported in 1991, 2005/2006 and 2025 [17–19]. Biodiversity and succession of communities have been studied extensively in this region, with main drivers of community succession including species tolerance to geochemical changes in vent fluids, biotic interactions as well as dispersal and larval supply [8,15,20]. Resistance to acute physiological stress is thus an important underlying factor influencing succession at hydrothermal vents but little studied.

The impact of any physiological stress depends on its intensity and duration. Thermal tolerance can therefore be quantified in either temperature or time units, conventionally defined as the median temperature or exposure duration at which 50% of individuals from a species collapse [21]. Both parameters have been sometimes called $LT_{50}$ (either lethal median time or lethal median temperature) [22,23]. In this study, $LT_{50}$ will be used to indicate lethal median time, and $LD_{50}$ (lethal median dose) for temperature. Another common measurement of thermal tolerance is the critical thermal maximum ($CT_{max}$), defined as the highest temperature a species can survive before the onset of muscle spasms or death during a gradual temperature increase [24].

No hydrothermal vent species seems to possess the physiological thermal tolerance to withstand temperatures exceeding 55°C for more than 2 h exposures, a

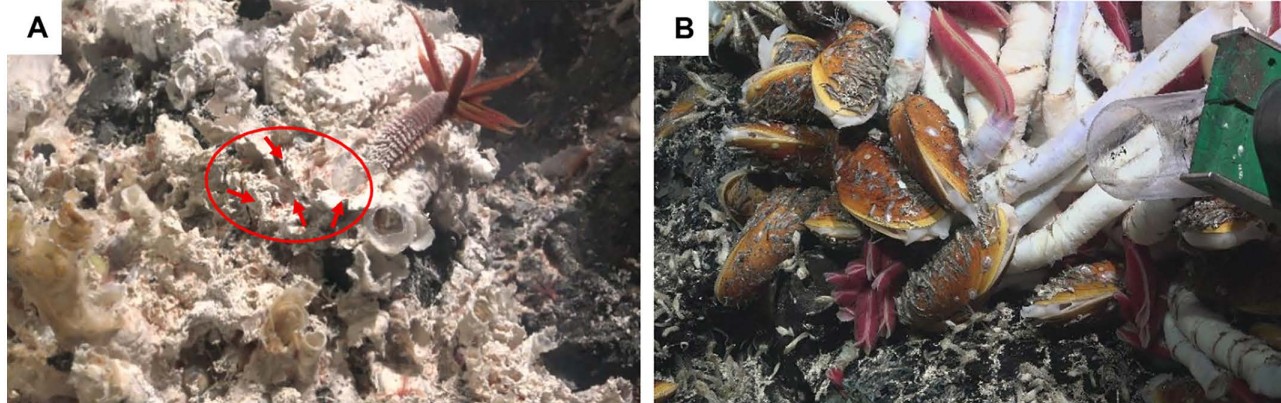

**Fig 1. Habitats of hydrothermal vent copepods.** A: Pompeii worm aggregation in the vicinity of focused vent flow, with an ~ 10 cm long *Alvinella pompejana* individual in the centre, and visible dirivultid copepod communities (red dots inside circle indicated by arrows) on the East Pacific Rise (9.840°; −104.292°) at ~2500 meters depth (Schmidt Ocean Institute); B: Tubeworm & mussel aggregation growing on and near diffuse flow emerging from basalt cracks, composed of the giant tubeworm *Riftia pachyptila* (*Riftia*'s plume ~10 cm long) and mussel *Bathymodiolus thermophilus* on the East Pacific Rise (9.840°; −104.292°) at ~2500 meters depth (Schmidt Ocean Institute). Copepod communities reside on top of tubeworm tubes and mussel shells, but their precise locations along these aggregations are still unclear.

value known as the metazoan upper thermal limit (UTL) [25]. For example, *Riftia pachyptila* and *Alvinella pompejana* for 2 h exposures reach 100% mortality in the range of 32–35°C and 50–55°C respectively [26,27]. Large mobile fauna, such as the decapod crustaceans *Rimicaris exoculata* and *Bythograea thermydron*, show 100% mortality at 38.5°C and 37.5°C [28,29]. Many vent animals, instead of reaching very high thermal tolerances, only possess resistance to the short-term heat fluctuations encountered in the turbulent mixing of vent fluid with bottom waters and have developed behavioural adaptations such as heat avoidance or recirculation of ambient water around the surface of their bodies [27,30,31].

High temperature can also accelerate metabolic rates and oxygen consumption in an already hypoxic environment, in a phenomenon known as Oxygen- and Capacity-Limited Thermal Tolerance (OCLTT) [32]. Several invertebrates, including vent species, have adapted to possess large quantities of high-affinity oxygen-binding proteins such as haemoglobin either for transport or storage to compensate for fluctuations in temperature, oxygen availability and consumption [33], allowing survival for short bursts or prolonged exposures to anoxia [34]. For example, *Riftia pachyptila* can survive anoxia up to 60 h at a temperature of 15°C, while the large vent clam *Calyptogena magnifica* at a similar temperature (14°C) can survive for a minimum of 16 h [34].

The physiological investigation of hydrothermal vent fauna poses significant logistical and methodological challenges unique to this extreme environment. The remote location of many deep-sea sites grants accessibility only for short intermittent periods of time (research expeditions), and with highly specialized equipment (ROVs or submersibles), limiting the number of possible replicates [35]. The presence of rapidly fluctuating physico-chemical regimes (e.g. temperature, oxygen, sulphide), alongside the elevated pressure difference experienced by animals when brought to the surface hamper recreation of near-realistic vent habitat conditions in a laboratory setting and rearing of selected species for long-term experiments [36]. Research efforts have focused on charismatic megafauna and macrofauna species, since their large size makes them easier to sample, identify and monitor visually compared to smaller species that require careful observation of static individuals through a microscope for the same tasks. Thus, empirical physiological data on meiofauna (size between 32 μm and 1 mm) [37] associated with vent mega/macrofauna communities, which have less possibilities to avoid or modify heat due to their small body size, has remained lacking despite being estimated to contribute up to 50% of the total diversity of deep-sea vents [14,38].

Copepods represent one of the most important meiofauna groups at deep-sea hydrothermal vents. Most species (78) belong to the family Dirivultidae (order Siphonostomatoida) and are only reported from vent ecosystems [1,39,40]. Siphonostomatoid copepods share a siphon-like mouth structure adapted for parasitic feeding, but dirivultid copepods differ from other families by using this structure to feed on fine grained food mainly composed of bacteria [41] (Fig 1, S1 Fig). Diets have been shown to be species-specific and range from exclusively autotrophic bacteria in focused flow habitats to a mixture of autotrophic and heterotrophic bacteria in diffuse flow habitats [42]. Several dirivultid copepod species exhibit high hemoglobin concentrations, potentially allowing them to uptake oxygen and to thrive in hypoxic or anoxic vent environments [43,44]. Contrary to most mega- and macrofauna, dirivultid copepods and other meiofauna are distributed across basalt habitats proximate and distant to vents. Not being restricted to vent flow suggests a broad physiological tolerance to temperature and oxygen concentration [14,45].

Physiological studies have historically focused on determining tolerance limits of single species, but more recently community approaches have been considered. Studies on terrestrial invertebrates living in a shared habitat [46,47] have already uncovered community-wide thermal tolerance trends that shift in response to temperature changes, either due to acclimation plasticity or rapid natural selection in short-lived species. In particular, Bujan et al., 2020 [46] have shown that in a comparison of several North American ant species from different genera the best predictor for critical thermal maximum ($CT_{max}$) is habitat type, with subterranean communities consistently having lower tolerances than ground or canopy communities. These shared community-wide trends appear to be broadly applicable across seasons or latitude to ectotherms, including copepods [48]. Copepod species inhabiting East Pacific Rise 9°50'N vent habitats belong to very few closely related genera from the same family. Their shared ecological niche and trophic level as small bacterial feeders can further justify a community approach, since every species in the same habitat needs to tolerate very similar physico-chemical conditions.

In this study, we investigate and compare the thermal tolerance of dirivultid copepod communities from the East Pacific Rise 9°50'N hydrothermal vent region in relation to habitat, exposure time, oxygen concentration and pressure by determining their respective $LT_{50}/LD_{50}$ and TDT survival curves. Dirivultid copepod communities from two habitats with different vent fluid regimes are investigated: Pompeii worm aggregations (focused flow) and tubeworm & mussel aggregations (diffuse flow). The objectives of our work were 1) to evaluate how thermal regimes in different habitats (focused flow at Pompeii Worms/diffuse flow at tubeworms & mussels) influence copepod's $LT_{50}/LD_{50}$ at different time scales, 2) to investigate whether the anoxic nature of vent fluids and the exposure to decreased pressure during sampling impact copepod survival, and 3) to discuss how the physiology of copepods may relate to their observed biodiversity and succession patterns at the studied location over the past decades.

## Materials and methods

### Study areas and *in situ* sampling

Hydrothermal vent copepods were collected at 9°50'N 104°W on the East Pacific Rise (EPR) at ~2500 meters depth (S1 File, Sample Sheet). The main study site was Tica vent, with 38 community samples. In addition, to enhance the number of replicates, five samples were opportunistically collected from the nearby Bio9 and three from Biovent, which were observed to contain the same habitat types. Sampling was conducted on board R/V Falkor (too) during 18 dives of ROV SuBastian within the framework of the "Underworld of Hydrothermal Vents" project [49] in July of 2023. Samples from surface Pompeii worm and surface tubeworm & mussel aggregations were collected with the ROV's suction sampler or by removing and storing megafauna inside bioboxes with the ROV's arm. After recovery on board of the ship, the water content of the bioboxes and suction samples was sieved through a 1000 μm and 32 μm mesh to separate the macrofauna and meiofauna fractions. Samples were stored in cold filtered seawater inside a cold room (6°C), and the meiofauna fraction was sorted using a Leica EZ4 W dissection microscope while kept cool on an ice tray to manually separate the

dirivultid copepods from the rest of the meiofauna sample. Dirivultid copepods were stored in seawater at 4°C inside shallow petri dishes to maximise oxygen exchange between water and atmosphere, until they were randomly selected for each incubation.

## Experimental setup

To recreate deep-sea conditions, incubations were performed inside pressure vessels [50]. The vessels, capable of holding three vials of 5.9 ml (where the animals of each replicate were housed), were pressurized using a Waters 515 HPLC Pump up to 200 bar and placed in a water bath (ARGOLAB WB 12 Lt) to keep a constant temperature. The Exetainer 719W vials used in the experiments were equipped with an airtight pressure relief cap, allowing pressure to be equalised inside the vial.

The static thermal tolerance method (quantified as the time of collapse at a constant stressful temperature) was chosen to lessen the frequency of depressurization during the experiment to check for survival [51]. The use of a sealed pressure vessel, alongside the minute size of copepods, meant that survival could only be assessed once the vessel was depressurized and opened. Death was chosen as the indicator for "collapse" since other effects such as loss of motor coordination, or the onset of muscle spasms would be challenging to quantify in meiofauna species [21]. Specimens were considered dead when, placed under a stereomicroscope, no spontaneous movement was detected, and individuals did not react when touched with a needle.

For vent copepods, four exposure durations (2, 4, 8, 10 hours) were selected, and for each a maximum of 8 temperatures were investigated (S1 Table). Temperatures were selected by first consulting the *in situ* ROV temperature probe as reference and then adapted to obtain a complete survival curve. To better represent the rapid temperature changes typical of vents, incubations were not preceded by an acclimation period. For every exposure duration, a control temperature of 4°C was included as the closest temperature to mimic the effect of the 1.9–2.1°C ambient bottom water measured at the Tica vent area [49] on copepod survival. Every incubation included 2 pressure vessels, pressurized at 3000 psi (200 bar, the closest pressure that could be achieved with the pressure vessels to the ~250 bar of our study site), with 3 replicate vials containing 10 copepods each. In the first pressure vessel, the vials were filled with filtered oxygenated seawater ("oxic vials"), whereas in the second vessel the vials contained anoxic seawater obtained by bubbling $N_2$ gas while monitoring oxygen levels with an oximeter dipping probe ("anoxic vials"). No sulfide was added to reach anoxia. The oxygen content of oxic and anoxic vials was measured before and after the incubations using adhesive sensor spots mounted on the inner wall of the incubation vial (Presens Self-adhesive Oxygen Sensor Spot SP-PSt3-SA) and a fibreoptic oximeter to control for leaks in the vials or anomalies in oxygen consumption. The Fiberoptic Oxygen Meter Fibox 4 was utilized for both types of oxygen measurements. To test for the effect of rapid pressurization/depressurization of the vessel on copepod survival, each 2 h exposure included one more set of 3 oxic and 3 anoxic replicate vials incubated at atmospheric pressure. Only 2 h exposures included pressure controls since the effect of a change in pressure would be more noticeable compared to longer exposures. In total, 118 incubations were performed in different temperature, oxygen, and pressure conditions with 3 replicates each containing 10 copepods, accounting for a total of 3540 hydrothermal vent copepods (S1 File, Data Sheet). After every incubation, the vials were cleaned with a 0.1% HCl solution, the number of dead copepod specimens was noted, and dead specimens were preserved in 95% ethanol for identification.

## Copepod community analysis

Copepod community analysis was carried out on shore. Material collected in areas beyond national jurisdiction has been imported and exported from Panama to the Netherlands (Import and export permit to and from Panama, Ministerio de Desarrollo agropecuario (215392–215398), and Ministerio de Ambiente (PA-05-ARB-131–2023); import permit from Panama to Netherlands, NVWA (NVWA-0166861)). A Leica S APO stereomicroscope and microscopy needles were used to

transfer and mount all preserved copepods onto microscope slides using pure glycerine as a mounting medium. A Leica DM 1000 microscope was used for taxonomic identification of copepod species, following Boxshall and Halsey, 2004, Desbruyères et al., 2006, Gollner et al., 2010a, Humes and Dojiri, 1980, Humes, 1987, Humes, 1989a, Humes, 1989b, Humes and Lutz, 1994, Humes and Segonzac, 1998, Ivanenko et al., 2011 [38,39,52–59]. From the dead specimens preserved and identified after the experiments (2124 in total), all but 29 could be identified to species level. We further note that 48 *Rhogobius* sp. individuals could not be distinguished between *R. contractus* and *R. rapunculus* and were thus labelled as "*Rhogobius* sp. (*contractus*/*rapunculus*)". A complete taxonomic breakdown of all copepod specimens identified for each sample can be found in the S1 File, Copepod ID.

**Thermal tolerance analysis**

The data analysis was performed using R version 4.3.0 with packages from the MASS library (for advanced statistical functions, e.g., $LD_{50}$ calculation), car (for regression diagnostics), lmtest (for heteroscedasticity testing) and tidyverse (for data wrangling). Firstly, the median lethal temperature ($LD_{50}$) corresponding to each exposure time ($LT_{50}$) was estimated by fitting the following survival curve from Hourdez et al., 2002 [60] to the temperature and respective copepod survival data using Nonlinear Least Squares regression and isolating the theoretical temperature value for which survival is 50%.

$$p = \frac{P_1}{1 + \left(\frac{T}{P_2}\right)^{P_3}}$$

Where:

- $p$ is the copepod survival proportion.

- $T$ is the temperature of the incubation.

- $P_1$ is the initial plateau of the curve. Under ideal conditions $P_1$ reaches values of 100%, but unwanted stress in the experiment setting can result in lower survival, which this parameter accounts for.

- $P_2$ is the $LD_{50}$ of the survival curve.

- $P_3$ is the slope of the curve.

When organisms are exposed to stressful temperatures, thermal tolerance decreases linearly with the logarithm of elapsed time, following a mathematical relation known as the Thermal Death Time (TDT) curve. Once $LD_{50}$ was extrapolated for each $LT_{50}$ exposure, a TDT curve could be generated by linear regression analysis using the following relationship [21]:

$$T = CT_{max} + z \cdot \log_{10} t$$

Where:

- $T$ is the $LD_{50}$ temperature.

- $CT_{max}$ is the $LD_{50}$ temperature resulting in death at $\log_{10} t = 0$, thus the maximum temperature of organismal functions failure.

- $z$ is the slope of the curve.

- $t$ is the $LT_{50}$ exposure time.

Curve fitting was considered of good quality for $R^2$ values higher than 0.7 and of acceptable quality for values between 0.6 and 0.7. Model adequacy was assessed through residual analysis, including tests for normality (Shapiro-Wilk test) and homoscedasticity (Breusch-Pagan test). After confirming normality assumptions (normality: $p > 0.05$; homoscedasticity: $p > 0.05$), Welch's *t*-tests were performed between the TDT slopes and intercepts of the two vent habitats to determine the statistical significance of the observed differences. Comparisons between $LD_{50}$ values of oxic/anoxic and pressurized/unpressurized 2 h survival curves for each habitat contained residuals that violated normality or homoscedasticity assumptions. In these cases, wild bootstrap resampling was employed as a robust alternative [61]. Wild bootstrap preserves the fitted model structure while resampling residuals, ensuring a realistic estimate of $LD_{50}$ in all resamples. P-values were calculated from 2000 iterations, providing distribution-free inference robust to non-normal error structures commonly observed in thermal tolerance experiments and allowing to determine the statistical significance of the effect of anoxia and pressure on copepod survival.

## Results

### Copepod community composition

The identification of over 1200 copepod specimens revealed that the community inhabiting the Pompeii worm habitat was almost entirely composed of the species *Stygiopontius hispidulus* (91.51%) alongside small contributions from individuals of *Stygiopontius appositus* (5.19%), *Ceuthoecetes acanthothrix* (1.24%) and other occasional species, found in numbers smaller than 10 individuals each, belonging to the genera *Aphotopontius*, *Nilva* and *Stygiopontius* (Fig 2, S2 Table). Almost every sample was from the Tica vent area, but the three samples from the nearby Bio9 site also consisted almost entirely of *Stygiopontius hispidulus*. Overall, the genus *Stygiopontius* made up more than 97% of the entire community. A single individual from a non-dirivultid species (*Tisbe* sp. 1) was discovered in the community, but its accidental inclusion is assumed not to have impacted our results significantly.

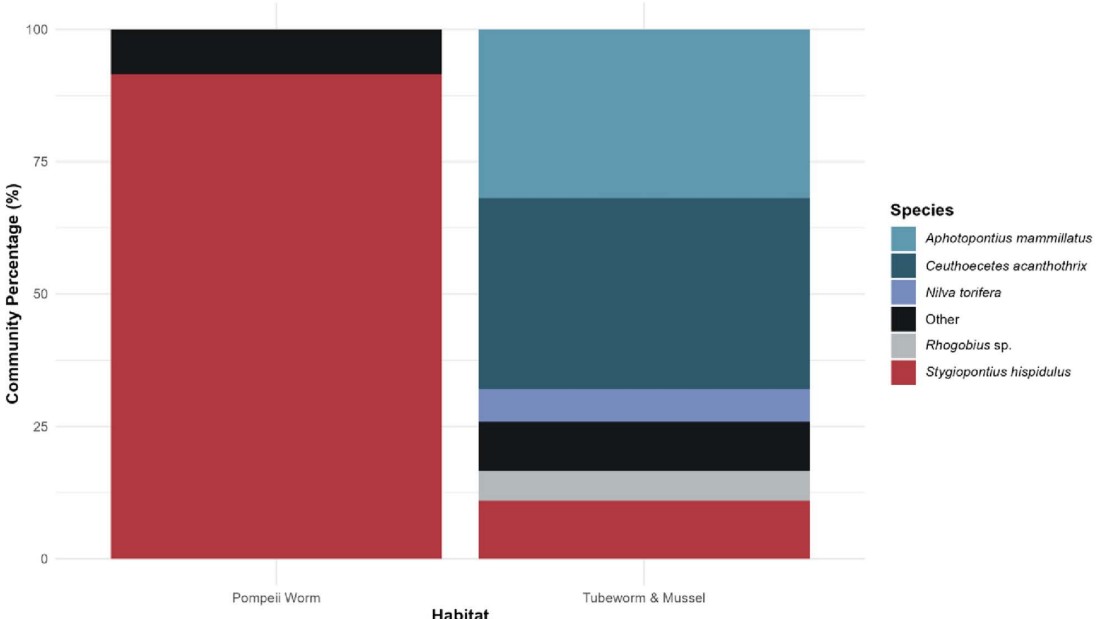

**Fig 2. Copepod species distribution from Pompeii worm and Tubeworm & mussel habitats.** In the tubeworm & mussel habitat, the category *Rhogobius* sp. refers to the combined abundance of all three species belonging to this genus. "Other" species category contains both dirivultid copepod species present in low densities and copepods that could not be identified to species level (for more information see S2 Table).

The identification of over 900 copepod specimens showed that tubeworm & mussel habitats hosted a much more diverse dirivultid copepod community. The most abundant dirivultid species were *Ceuthoecetes acanthothrix* (36.00%), *Aphotopontius mammillatus* (31.94%), *Stygiopontius hispidulus* (10.98%) and *Nilva torifera* (6.15%). Smaller contributions came from several species belonging to the genera *Aphotopontius*, *Ceuthoecetes*, *Exrima*, *Rhogobius*, *Scotoecetes* and *Stygiopontius* (Fig 2, S2 Table). Almost every sample was from the Tica vent area, but the three samples from the nearby Biovent site showed a majority of *Aphotopontius mammillatus*. Only two individuals from non-dirivultid species (*Halectinosoma* sp. 1 and *Idomene* sp.1) were discovered in the community, but their accidental inclusion is assumed not to have impacted our results in any significant way.

## Thermal tolerance at *in situ* pressures under oxic conditions

Oxygen concentration was on average $163.88 \pm 5.82$ µM before and dropped to $95.63 \pm 9.14$ µM after the incubations, likely due to respiration. It was observed that in 11 out of 147 oxic pressurized vials, despite starting with standard surface water oxygen concentrations, the oxygen level dropped close to hypoxic (2 mg/l, or 62.50 µM) [62] or anoxic conditions by the end of the incubation ($0,67 \pm 0,09$ µM on average) but mortality was not affected appreciably.

Under short 2 h exposures, dirivultid copepods associated with Pompeii worm habitats exhibited the greatest thermal tolerance with a $LD_{50}$ of $39.8 \pm 0.4$°C (Fig 3A, Table 1). Pompeii worm copepods also registered the highest temperature at which any individuals survived (two vials with one live copepod each), being 43°C for 2 h. Increasing exposure time to 8 h, $LD_{50}$ dropped drastically (>15 °C change) to $23.1 \pm 0.2$°C. The copepod community of tubeworm & mussel habitats, in contrast, had a much lower $LD_{50}$ for 2 h exposures ($32.4 \pm 0.9$°C) but a similar 8 h $LD_{50}$ of $25.4 \pm 0.4$°C (Fig 3B, Table 1).

All survival curves exhibited good or acceptable fitting apart from the 4 h curve for tubeworm & mussel copepod species, which demonstrated the lowest $R^2$ of 0.457, and apart from the 10 h $LT_{50}/LD_{50}$ survival curve for Pompeii worm copepods that was unable to generate a fitting due to a lack of data points along the curve's flex point, causing an error signal in the nonlinear least squares analysis (Table 1). The initial plateau of each survival curve appears to reach a proportion of less than 1 since control temperatures exhibit a certain level of background mortality, accounted for by the parameter $P_3$ of the nonlinear model. Both copepod communities display a 2 h survival curve markedly distinct from longer exposures in terms of thermal tolerance (Fig 3).

Estimates of $CT_{max}$ ($LD_{50}$ temperature resulting in death at $\log_{10} t = 0$, thus the maximum temperature of organismal functions failure) are $46.9 \pm 4.7$°C for Pompeii worm copepods and $36.4 \pm 5.2$°C for tubeworm & mussel copepods (Table 2). The thermal death time curves (TDT, Fig 4) of both copepod communities in oxic conditions and habitat pressure have a steep slope value ($-27.8 \pm 7.2$ for Pompeii worm copepods and $-16.5 \pm 6.9$ for tubeworm & mussel copepods), owing to the distinct drop in survival for exposures longer than 2 h. Both curves exhibit good fitting, having $R^2$ values of 0.734 or higher, both follow a normal distribution (p-values 0.921 and 0.517 for Pompeii worm and tubeworm & mussel copepods respectively) and are homoscedastic (p-value 0.380). When compared through a *t*-test no statistically significant difference in slope or intercept between the two was observed (p-values 0.353 for slopes and 0.235 for intercepts).

## Thermal tolerance at *in situ* pressure and anoxic conditions

Under short (2 h) exposures and anoxic conditions (average of $0.24 \pm 0.11$ µM and $1.76 \pm 0.38$ µM oxygen before and after the experiments, respectively), $LD_{50}$ was $26.9 \pm 0.9$°C for Pompeii worm copepods and $27.8 \pm 0.9$°C for tubeworm & mussel copepods, a 12.9°C and 4.6°C drop respectively compared to oxic conditions (Fig 5, Table 1). For longer exposures, survival was often too low to construct a curve. Only the 2 h survival curve could be fitted to the Pompeii worm copepods' anoxic survival data ($R^2$ of 0.704) due to a large increase in mortality, making it impossible to carry out a comparison of TDT curves. Instead, a comparison between 2 h $LT_{50}/LD_{50}$ survival curves under oxic and anoxic conditions was carried out. Due to the non-normal distribution of the anoxic survival curve data, this comparison was carried out by bootstrapping (p-value 0.130; not significant). Similarly, the 10 h survival curve of tubeworm & mussel copepods could not be fitted due

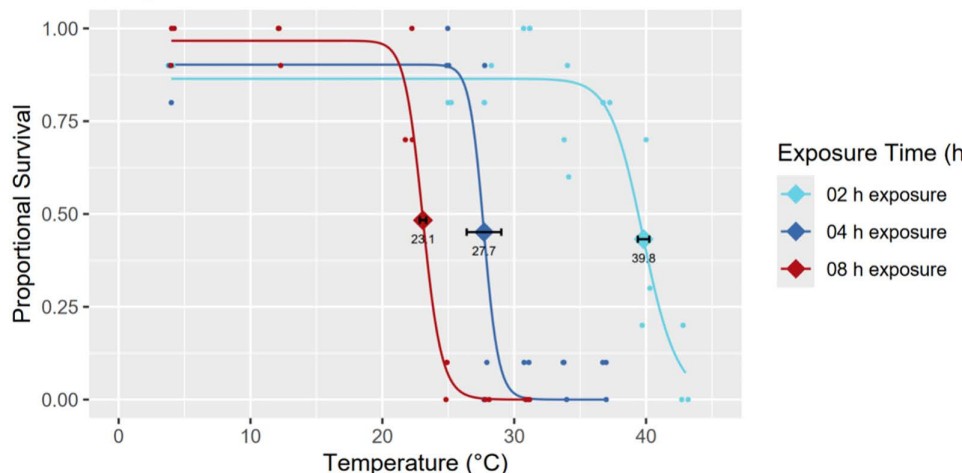

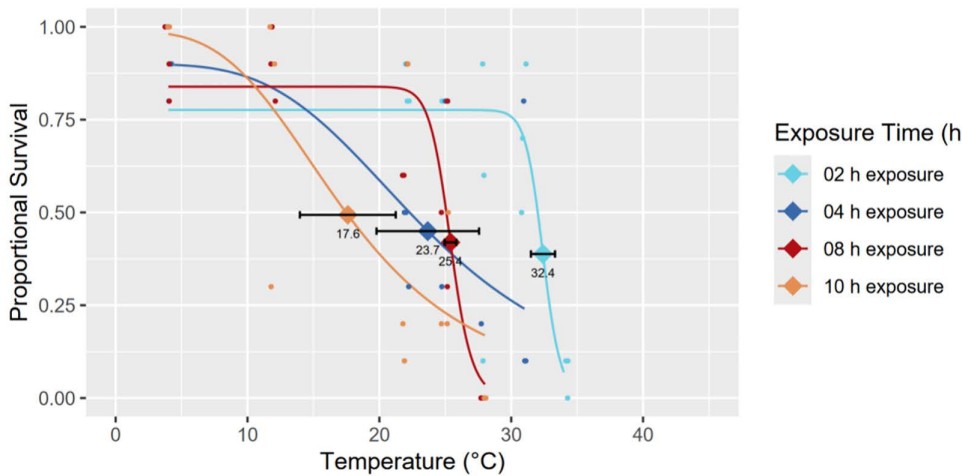

**Fig 3. LT$_{50}$/LD$_{50}$ survival curves of copepod communities under oxic conditions at ambient habitat pressure under different exposure times.** Copepod data is shown from Pompeii worm (A) and tubeworm & mussel (B) habitats, pressurized at 200 bar. For each survival curve, the respective LD$_{50}$ value with standard error bars is marked with a diamond shape. Proportional survival given in %, temperature in °C. (A) Pompeii worm copepods have a LD$_{50}$ of 39.8±0.4°C for 2h exposures, 27.7±1.3°C for 4h exposures and 23.1±0.2°C for 8h exposures. (B) Tubeworm and mussel copepods have a LD$_{50}$ of 32.4±0.9°C for 2h exposures, 23.7±3.9°C for 4h exposures, 25.4±0.4°C for 8h exposures and 17.6±3.6°C for 10h exposures. For curve fitting statistics see Table 1.

to high mortality at temperatures higher than 4°C. The remaining curves show good fit to the data ($R^2$ of 0.779 and 0.894 for 2 and 4h respectively), apart from the 8h exposure ($R^2$ of 0.278) which also presents a large standard error margin (Table 1). The data for the resulting TDT curve is normally distributed (Shapiro-Wilk test p-value 0.979) and homoscedastic (p-value 0.375). *T*-tests between the TDT curves of tubeworm & mussel copepods under oxic and anoxic conditions

**Table 1. Thermal tolerance and curve fitting statistics for all LT$_{50}$/LD$_{50}$ survival curves of tubeworm & mussel (T&M) and Pompeii worm (Pompeii) copepod communities.**

| Habitat | [O$_2$] | Pressure (bar) | Exposure (h) | LD$_{50}$ (°C) | SE | p-value | $R^2$ |
|---|---|---|---|---|---|---|---|
| Pompeii | Oxic | 200 | 2 | 39.8 | 0.4 | < 2· 10$^{-16}$ | 0.840 |
| Pompeii | Oxic | 200 | 4 | 27.7 | 1.3 | 1.44· 10$^{-12}$ | 0.813 |
| Pompeii | Oxic | 200 | 8 | 23.1 | 0.2 | < 2· 10$^{-16}$ | 0.978 |
| Pompeii | Oxic | 1 | 2 | 32.0 | 1.8 | 3.21· 10$^{-15}$ | 0.610 |
| Pompeii | Anoxic | 200 | 2 | 26.9 | 0.9 | 2.83· 10$^{-16}$ | 0.704 |
| Pompeii | Anoxic | 1 | 2 | 24.9 | 0.2 | < 2· 10$^{-16}$ | 0.946 |
| T&M | Oxic | 200 | 2 | 32.4 | 0.9 | 7.14· 10$^{-16}$ | 0.639 |
| T&M | Oxic | 200 | 4 | 23.7 | 3.9 | 5.43· 10$^{-5}$ | 0.457 |
| T&M | Oxic | 200 | 8 | 25.4 | 0.4 | 5.24· 10$^{-16}$ | 0.839 |
| T&M | Oxic | 200 | 10 | 17.6 | 3.6 | 4.03· 10$^{-5}$ | 0.663 |
| T&M | Oxic | 1 | 2 | 30.0 | 0.3 | < 2· 10$^{-16}$ | 0.878 |
| T&M | Anoxic | 200 | 2 | 27.8 | 0.9 | 1.11· 10$^{-14}$ | 0.779 |
| T&M | Anoxic | 200 | 4 | 23.4 | 0.5 | 1.49· 10$^{-14}$ | 0.894 |
| T&M | Anoxic | 200 | 8 | 19.2 | 8.5 | 0.0427 | 0.278 |

SE (standard error) and p-value refer to the LD$_{50}$ temperature extrapolated from each survival curve. $R^2$ refers to the predictive accuracy of the survival curve with respect to the copepod survival data.

**Table 2. Fitting statistics for TDT Curves under oxic and anoxic conditions at habitat pressures.**

| Habitat | CT$_{max}$ (°C) | SE | p-value | Slope | SE | p-value | $R^2$ |
|---|---|---|---|---|---|---|---|
| Pompeii oxic | 46.9 | 4.7 | 0.0631 | −27.8 | 7.2 | 0.1608 | 0.937 |
| T&M oxic | 36.4 | 5.2 | 0.0202 | −16.5 | 6.9 | 0.1410 | 0.738 |
| T&M anoxic | 32.1 | 0.1 | 0.0021 | −14.4 | 0.2 | 0.0070 | 0.999 |

CT$_{max}$ is equivalent to the intercept of the TDT curve. SE (standard error) and p-value refer to the CT$_{max}$ and slope from each TDT curve. $R^2$ refers to the predictive accuracy of the survival curve with respect to the copepod survival data.

resulted in high p-values for slopes and intercepts of 0.786 and 0.505 respectively (see S2 Fig for plot of the TDT curve and Table 2 for fitting statistics).

## Thermal tolerance at atmospheric pressure

Reduction in survival was not statistically significant when deep-sea copepods were incubated at atmospheric pressure. Compared to their pressurized counterparts, the temperature tolerance shifts in oxic conditions ranged from almost negligible (from 32.4 ± 0.5 to 30 ± 0.3°C, bootstrap p-value of 0.075) for tubeworm & mussel copepods to large (from 39.8 ± 0.4 to 32 ± 1.8°C, bootstrap p-value of 0.106) in Pompeii worm copepods over 2 h in oxic conditions (Fig 6, Table 1). In anoxic conditions, comparisons between the 2 h survival curves of anoxic pressurised and unpressurised specimens could only be carried out with Pompeii worm copepods (see S3 Fig) due to high mortality in the equivalent tubeworm & mussel community. LD$_{50}$ of the anoxic unpressurized survival curve was 24.9 ± 0.2°C, a slight drop compared to the 26.9 ± 0.9°C of the anoxic pressurized curve. However, bootstrapping revealed no significant difference between the two curves (p-value = 0.053) despite good $R^2$ values for both survival curves (0.704 and 0.946 respectively).

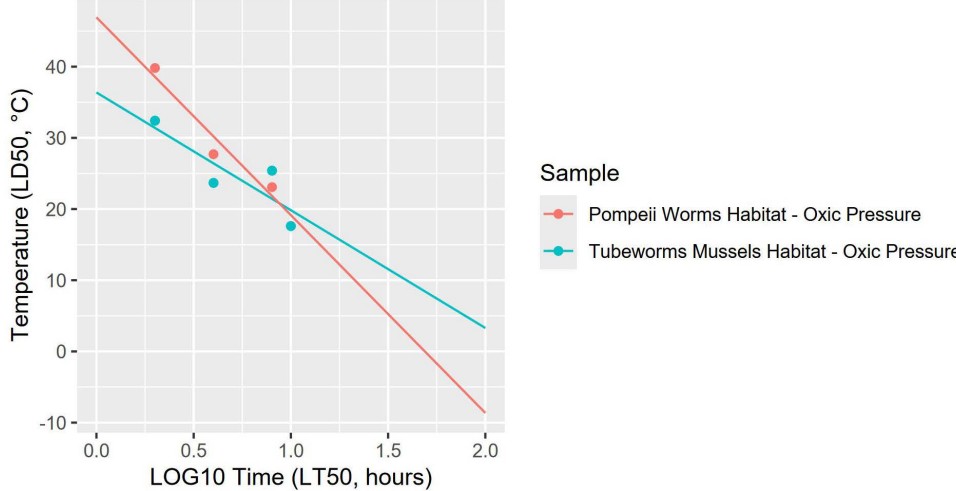

**Fig 4. Thermal Death Time (TDT) curves of copepod communities under oxic conditions and ambient habitat pressure.** Incubations were performed at 200 bar. Points represent $LD_{50}$ values extrapolated from the survival curves. Pompeii worm copepods (red line) have $CT_{max}$ of 46.9±4.7°C and slope −27.8±7.2, while tubeworm & mussel copepods (blue line) have $CT_{max}$ of 36.4±5.2°C and slope −16.5±6.9. For curve fitting statistics see Table 2.

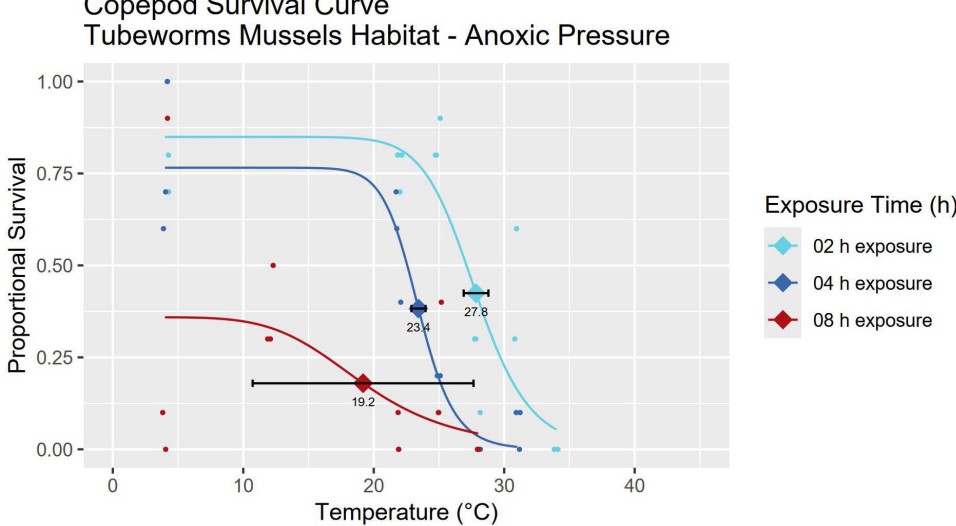

**Fig 5. $LT_{50}/LD_{50}$ survival curves for tubeworm & mussel copepods under anoxic conditions at ambient habitat pressure.** Incubations were performed at 200 bar. For each survival curve, the respective $LD_{50}$ value with standard error bars is marked with a diamond shape. Tubeworm & mussel copepods under anoxic conditions have a $LD_{50}$ of 27.8±0.9°C for 2 h exposures, 23.4±0.5°C for 4 h exposures and 19.2±8.5°C for 8 h exposures. For curve fitting statistics see Table 1.

## Discussion

In this study, we investigated the thermal tolerance of dirivultid copepod communities from two different hydrothermal vent habitats along EPR 9°50'N under different oxygen and pressure conditions. The focused flow copepod community was almost exclusively composed of a single species (*Stygiopontius hispidulus*). The diffuse flow community was more diverse

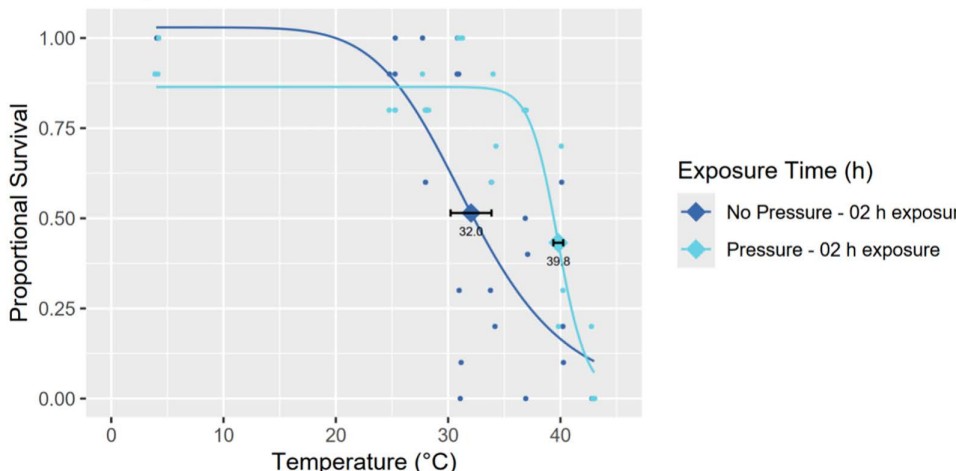

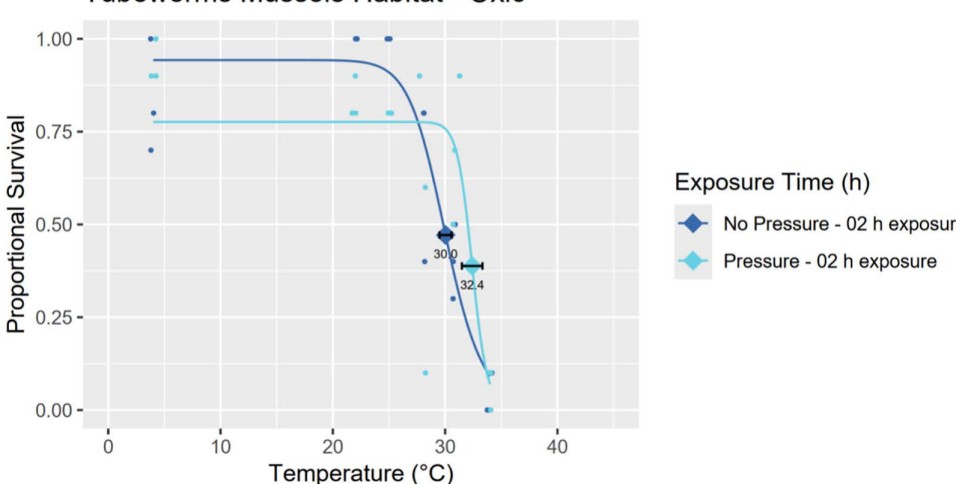

**Fig 6. LT$_{50}$/LD$_{50}$ survival curves of copepod communities under oxic conditions at ambient habitat pressure (200 bar) and atmospheric pressure.** Copepods are from Pompeii worm (A) and tubeworm & mussel (B) habitats. For each survival curve, the respective LD$_{50}$ value with standard error bars is marked with a diamond shape. (A) Pompeii worm copepods at *in situ* pressure have a LD$_{50}$ of 39.8±0.4°C for 2h exposures and at atmospheric pressure have a LD$_{50}$ of 32.0±1.8°C for 2h exposures. (B) Tubeworm & mussel copepods at *in situ* pressure have a LD$_{50}$ of 32.4±0.9°C for 2h exposures and at atmospheric pressure have a LD$_{50}$ of 30.0±0.3°C for 2h exposures. For curve fitting statistics see Table 1.

while still mainly dominated by two species (*Ceuthoecetes acanthothrix* and *Aphotopontius mammillatus*). Overall, dirivultid copepods appear resistant to high temperatures only for short exposures, with the most tolerant community being from the focused flow, high temperature Pompeii worm habitat. Survival was negatively impacted by anoxia, whereas atmospheric pressure did not cause a statistically significant drop in survival.

## Copepod survival under oxic conditions at *in situ* pressure

Results under oxic conditions at habitat pressure demonstrate that copepods are capable of remarkable thermal tolerance only for short exposure times. The short-time temperature tolerance is consistent with thermal regimes present at vents, where turbulent mixing of warm vent fluids and cold bottom water generates steep thermal gradients and rapid temperature fluctuations [10]. Consequently, mobile vent fauna needs to be equipped to resist these unpredictable changes while seeking cooler fluids in a complex three-dimensional environment [30]. The 2 h thermal tolerance of the Pompeii worm copepod community (39.8±0.4°C), mostly composed of *Stygiopontius hispidulus*, is one of the highest experimentally recorded for vent fauna [27,63]. Pompeii worm copepods are more temperature tolerant than tubeworm & mussel copepods (2h LD$_{50}$ 39.8±0.4°C/ 32.4±0.9°C and CT$_{max}$ 46.9±4.7°C/ 36.4±5.2°C respectively), in line with the observed difference in vent fluid between the two habitats, which is more extreme in the focused flow Pompeii worm habitat [8,10].

Dirivultid copepods that reside on the barren outer surface of the Pompeii worm tubes (Fig 1A), are exposed to unstable thermal conditions with high temperature peaks, requiring high CT$_{max}$. Vent fluid temperatures on top of *Alvinella* tubes in the Tica vent area typically hover between 10 and 30°C, with occasional short-lived spikes above 40°C and frequent exposures to the 2°C ambient bottom water due to turbulent mixing [9]. The extreme environmental conditions at Pompeii worm habitats thus require very high thermal tolerance but very little tolerance to anoxia for the associated fauna. The CT$_{max}$ of Pompeii worm copepods (46.9±4.7°C) aligns just above the expected temperature peaks at focused flow habitats, whilst the CT$_{max}$ of tubeworm & mussel copepods (36.4±5.2°C) lies below them. Thus, the distinct CT$_{max}$ values observed for copepods from Pompeii worm and tubeworm & mussel habitats indicate that physiological limits drive community composition at the Pompeii worm habitat.

On the other hand, dirivultid copepods from tubeworm & mussel habitats possess a CT$_{max}$ (36.4±5.2°C) that is much higher than the highest vent fluid temperatures recorded amongst tubeworms & mussels, mirroring the broad ecological niche they can inhabit *in situ*. From the 23 dirivultid copepod species identified in studies at the 9°N EPR habitats, only three dirivultid species have been reported as vent specialists, i.e., being reported from only one vent habitat type [20]. Notably, diffuse flow temperatures have been shown to vary significantly even in neighbouring fauna aggregations, sustained by a shared underground hydrothermal reservoir, depending on the mixing proportion of undiluted vent fluid and seawater [10,13]. Thus, the CT$_{max}$ of ~36°C may be required to maintain populations within vent fields.

## Copepod survival under anoxic conditions at *in situ* pressure

Pompeii worm copepods, namely the species *Stygiopontius hispidulus* which highly dominates the community (Fig 2), were not able to survive high temperatures under anoxia for prolonged periods. This finding is in accordance with the natural conditions at Pompeii worm habitats, where temperature and oxygen concentration highly fluctuate within seconds at small spatial scales, but do not occur over prolonged periods [7]. Further, the surface of the chimneys is characterized by comparatively high oxygen concentration in relation to temperature because of conductive heating by the chimney wall [9]. We note that our experimental set-up simulated anoxia (water was bubbled with N$_2$), but we did not simulate the presence of sulfide, a chemical present in all vent habitats and known to be toxic to aerobic eukaryotes [6]. Survival of copepods under anoxia and in the presence of sulfide may be even lower than what we observed for anoxia alone. Prolonged survival under anoxic conditions at low temperatures (up to 10 h at 4°C) could be attributable to the low respiration rates recorded at low temperatures [64].

Low vent copepod survival rates under anoxic conditions and high temperatures align with estimates of structural and functional properties of haemoglobin from dirivultid copepods collected at the Juan de Fuca ridge. In the study by Hourdez et al., 2000 [43], haemoglobin with high oxygen affinity was found to represent about 60% of the total soluble proteins extracted from the dirivultid copepod *Benthoxynus spiculifer*. Despite being so abundant as to impart a red colour to the copepod (similarly to dirivultid copepod species in our study, see Fig 1A), it did not provide a significant storage pool of

oxygen, estimated to last less than 2 minutes at 15°C and about 30 seconds at 25°C. Consequently, we hypothesise *Stygiopontius hispidulus* and other less abundant Pompeii worm copepods to also possess a haemoglobin pool insufficient to allow more than short forays into warm anoxic microhabitats, and that anaerobic metabolism may be insufficient to keep up with increased energetic demand at high temperatures. Oxygen-depleted hydrothermal fluid, mixing turbulently with oxygen-rich bottom waters in the vicinity of vent openings, may create a habitat where oxygen rarely reaches concentrations close to saturation for prolonged periods [65]. Thus, dirivultid copepods likely express haemoglobin to maintain a concentration gradient from the outside to the inside of the animal when environmental oxygen concentrations are low but not anoxic, allowing them to continuously capture oxygen from the surrounding water [43,44].

In comparison, copepods inhabiting milder diffuse flows at tubeworm & mussel aggregations displayed greater tolerance to anoxia. If present in high density, tubeworms and mussels are known to modify and stabilize environmental conditions by trapping warm vent fluid inside their aggregations [13]. We speculate that the great megafauna density of tubeworm tubes and mussel shells with their dense byssus threads around diffuse flow emissions (Fig 1B) could provide less opportunities for rapid mixing between ambient oxygen-rich bottom waters and vent fluids [66,67]. This may create a less oxygenated environment for the copepods that live near the base of mussels and tubeworms, compared to the open surface of Pompeii worms where copepods crawl on (see copepods as red dots on Fig 1A). To test this hypothesis, the precise location of copepods within tubeworm & mussel aggregations as well as measurements of temperature, oxygen but also sulfide at the base of tubeworm & mussel aggregations would be needed.

## Copepod physiology and community succession at hydrothermal vents

At EPR 9°50'N, hydrothermal vent communities follow a cyclical pattern of colonization and disruption caused by periodic volcanic eruptions, with the latest eruptions having occurred in 1991–1992, 2005–2006 and 2025 [16–19]. Since 2001, diverse meiofauna community studies, including species-specific determination of dirivultid copepods, have been carried at Pompeii worm and tubeworm & mussel habitats of different ages [14,15,37] (see S1 File, Community compositions). Data on relative abundance of dirivultid copepods in their habitats over the past two decades, coupled with data on physiological limits of dirivultids, suggest that physiological limits strongly impact community composition at the Pompeii worm habitat regardless of successional stage. In contrast, complex interplays of interspecific competition, food-source partitioning and small-scale environmental heterogeneity within megafauna aggregations might shape dirivultid community dynamics in tubeworm & mussel habitats.

The dirivultid copepod communities were remarkably stable in the Pompeii worm habitat over the past decades (S1 File, Community compositions). The species *S. hispidulus* has always been the dominant species with >90% relative abundance, whether samples were taken pre- or post-eruption, in young or old habitats. The $CT_{max}$ of Pompeii worm copepods (46.9±4.7°C), aligns just above the expected temperature peaks at Pompeii worm habitats [9,10], whilst the $CT_{max}$ of tubeworm & mussel copepods (36.4±5.2°C) lies below them. In accordance with the natural experienced habitat conditions at Pompeii worms, anoxia likely is not a major constraint (see discussion above). Thus, the distinct $CT_{max}$ values observed for copepods from Pompeii worm habitats indicate that physiological temperature limits drive community composition at the Pompeii worm habitat.

The dominance of dirivultid species in the tubeworm & mussel habitat is highly variable over time (S1 File, Community compositions). The relatively high $CT_{max}$ and 2 h $LD_{50}$ of dirivultid copepods might mirror the broad thermal niche needed to sustain populations locally and over time in the tubeworm & mussel habitats. The $CT_{max}$ of 36.4±5.2°C is much higher than the measured temperatures of 11.6–24.9°C in smaller neighbouring tubeworm clumps at the time of sampling in 2023 [49] but is only slightly higher than the temperature of 32°C measured amongst tubeworms before the 2006 eruption [13]. However, it needs to be acknowledged that little is known on the experienced habitat conditions for copepods in the tubeworm & mussel habitat and how the ecological niches of each species impact their exposure to elevated temperature and low oxygen concentrations. Only more detailed abiotic measurements coupled with species-specific survival

curves would show if the thermal tolerances of each species represent a shared response akin to that of the broader community or if instead the community results are a weighted average of each species' thermal tolerance.

Competition and resource partitioning likely play important roles in structuring copepod communities over space and time at tubeworm & mussel aggregations. In this study, *Stygiopontius hispidulus* was found in both focused and diffuse flow habitats, but higher competition with other copepod species in tubeworm & mussel habitats, coupled with a greater acclimation potential, may lead to a preference of *S. hispidulus* for focused flow Pompeii worm habitats. Since post-eruption the dirivultid species *Ceuthoecetes acanthothrix* has undergone a significant increase, becoming the most abundant species in our community samples from 2023 and reaching values comparable to tubeworm collections from the year 2001/02 (S1 File, Community compositions). The specific mouth structure of this genus, a cutting borer, has been suggested to be related to parasitism associated to *Riftia pachyptila* in contrast to the typical bacterial feeder lifestyle of other dirivultid genera [58]. The lack of *Ceuthoecetes* in one to four-year-old tubeworm clumps mainly composed of *Tevnia* tubeworms [15], suggests that this species/genus might have a species-specific host association with *Riftia*. A study by Limén et al., 2007 [68] has shown that dirivultids occupy different feeding niches, with specialized bacterial and detritus feeders. To conclude, complex interplays of interspecific competition, food-source partitioning and small-scale environmental heterogeneity within megafauna aggregations might shape dirivultid community dynamics in tubeworm & mussel habitats.

### Vent fauna survival at atmospheric pressure

Tolerance to acute thermal stress at atmospheric pressure has not been studied extensively in deep-sea hydrothermal vent species, but observations have shown a wide range of responses to depressurization with varying degrees of severity. Vent copepods incubated under oxic conditions at atmospheric pressure in this study showed a decrease in $LD_{50}$ compared to their pressurized counterparts which was not statistically significant. Similarly, a combination of anoxia and atmospheric pressure did not yield a statistically significant drop in survival compared to anoxia at *in situ* pressure, suggesting there is no synergistic effect between oxygen concentration and pressure. Information on macro- and megafauna is more abundant, but far from exhaustive. Specimens of the deep-sea vent shrimp *Miocaris fortunata* collected at Rainbow vent (2300 m) displayed $LD_{50}$ at atmospheric pressure ranging from 21°C at 14h to 10°C for 36h, compared to the proposed $CT_{max}$ of 36±1°C at *in situ* pressures [29]. Other deep-sea crustaceans, such as *Rimicaris exoculata*, often appear dead after collection but resume normal activity when returned to a hydrostatic pressure typical of their habitat after a few minutes or hours [69,70]. Conversely, using aquaria that mimic the thermal and chemical conditions of hydrothermal vents at atmospheric pressure, an array of species including vent decapods *Gandalfus yunohana* and *Shinkaia crosnieri* have been successfully maintained for as long as one year, with certain species even molting, spawning and hatching in captivity [36]. Vent copepods retained normal movement when brought to atmospheric pressure and could be maintained alive on board the research vessel for a maximum of 3 days at 4°C without the use of specialized aquaria, suggesting high physiological tolerance to pressure variations (Messora, Gollner pers. obs.).

### Temperature and anoxia tolerance trends across vent community size classes

Broadening our scope from dirivultid copepods to the entire focused and diffuse flow habitat communities, the known upper thermal tolerances of meio-, macro-, and megafauna exposed to the same type of vent fluid are remarkably similar, while tolerance to hypoxia or anoxia seems to increase with size.

At focused flow habitats, the Pompeii worm copepod community, mostly composed of *Stygiopontius hispidulus*, is one of the highest experimentally recorded 2h thermal tolerances for vent fauna (39.8±0.4°C) but does not surpass the Pompeii worm *Alvinella pompejana* (50–55°C) [27,63] on which they reside [37] (Fig 1A). The barren outer surface of the Pompeii worm tubes leaves them exposed to turbulent and unstable thermal conditions, leading to high thermal tolerance but very little tolerance to anoxia (see discussion above). A similar situation has been recorded in vent scale

worms (macrofauna) from active chimney walls, with an upper thermal limit of 38°C and the triggering of active avoidance behaviour at low oxygen levels [63]. The greater tolerance of the Pompeii worm is likely due to their semi-sessile nature and closer proximity of the base of their tubes to focused flow vent openings [9,63].

Diffuse flow habitats, on the other hand, are typically characterized by large and dense forests of tubeworms and mussels (Fig 1B) that may act like a lattice that traps warm anoxic vent fluid at the center of the clump [13], reducing turbulent mixing with ambient water and stabilizing the environment. It is currently unknown exactly where copepods reside in this very complex habitat, but our $LD_{50}$ measurements in oxic ($LD_{50}$ 2 h 32.4±0.9°C) and anoxic ($LD_{50}$ 2 h 27.8±0.9°C) conditions suggest that they would be able to thrive virtually anywhere and would even be able to venture at the centre of the tubeworm clump for several hours in the diffuse flows measured at Tica vent before the most recent eruption. Diffuse flow megafauna, such as the crab *Bythograea thermydron* or the giant tubeworm itself *Riftia pachyptila* share a very similar upper thermal tolerance of 30–35°C for short exposures [26,28]. Anoxia, while barely survivable for 8 h by dirivultid copepods, can be tolerated for 12 h by the larger *B. thermydron* and between 36 and 60 h by *R. pachyptila* [34,71].

## Conclusions

In this study, a community-wide approach was employed to investigate the physiological limits of hydrothermal vent dirivultid copepods from focused and diffuse flow habitats to temperature, anoxia and pressure. Dirivultid copepods from hydrothermal vents possess high thermal tolerance exclusively for short exposures but have limited tolerance to long term exposures. The type of habitat, and consequently vent fluid flow, is an important predictor for maximum tolerated temperature since Pompeii worm copepods survived higher temperatures than tubeworm & mussel copepods. The upper thermal tolerances of dirivultid copepods are in the same range as macrofauna and megafauna from their respective habitat. Additionally, anoxia had a major negative impact on survival, especially on copepods from Pompeii worm aggregations, indicating that the primary role of haemoglobin may not be oxygen storage, but rather the creation of an oxygen gradient in hypoxic conditions. It is currently unknown if vent copepods may possess mechanisms such as glycogen storage to cope with anaerobic conditions. Endurance to anoxia is thus reduced in dirivultid copepods compared to macrofauna and megafauna from their respective habitat. Lastly, incubating copepods at atmospheric pressure resulted in no statistically significant reduction in survival and no synergistic effect when combined with anoxia. Copepod community diversity, as expected from previous studies [15], was low at focused flow habitats, mostly composed of a single species, but higher at diffuse flow habitats. These differences in community diversity, coupled with thermal tolerance determinations, suggest physiological constraints as the main driver of community composition at focused flow habitats and more complex biotic and abiotic interactions shaping dirivultid communities at diffuse flow habitats.

## Supporting information

**S1 Fig. Typical habitus of dirivultid copepods.** *Stygiopontius pectinatus* female SEM micrographs. A: ventral view. B: dorsal view. Scale bars 100 µm. Images originally from [38] and adapted in [45].
(TIF)

**S2 Fig. Comparison of tubeworm & mussel copepod TDT curves under oxic and anoxic pressurized conditions.** Incubations were performed at 200 bar. Points represent $LD_{50}$ values extrapolated from the survival curves. Tubeworm & mussel copepods in anoxic conditions (red line) have $CT_{max}$ of 32.1±0.1°C and slope −14.4±0.2. Tubeworm & mussel copepods in oxic conditions (blue line) have $CT_{max}$ of 36.4±5.2°C and slope −16.5±6.9. For curve fitting statistics see Table 2.
(TIF)

**S3 Fig. Comparison between anoxic Pompeii worm copepod incubations under pressurized and unpressurized conditions.** $LD_{50}$ values with standard error bars are marked with a diamond shape. Proportional survival given in %,

temperature in °C. Pompeii worm copepods at *in situ* pressure have a $LD_{50}$ of $26.9 \pm 0.9$°C for 2 h exposures and at atmospheric pressure have a $LD_{50}$ of $24.9 \pm 0.2$°C for 2 h exposures. For curve fitting statistics see Table 1.
(TIF)

**S1 Table. Temperature and exposure durations of all incubations performed on Pompeii worm and tubeworm & mussel (T&M) dirivultid copepod communities.**
(DOCX)

**S2 Table. Complete taxonomic breakdown of species identified in Pompeii worm and tubeworm & mussel copepod communities, the total number of individuals identified in each habitat and percentage over the whole community.** *Rhogobius contractus* and *R. rapunculus* could not be separated reliably and are thus labelled together as "*Rhogobius* sp. (*contractus*/*rapunculus*)".
(DOCX)

**S1 File. Supplementary materials and primary data.** This spreadsheet includes: Metadata, which offers a description of every column of the following four sheets; Data sheet, where the raw survival data is shown; Sample sheet, where details and origin of each copepod sample is shown; Copepod ID, where the taxonomic identification of each dead copepod specimen is shown; Community compositions, where data on the dirivultid copepod communities of both habitats collected from 2001 to 2023 is shown.
(XLSX)

## Acknowledgments

We thank the captain, crew, and marine technicians of *R/V Falkor (too)* and ROV SuBastian for their exceptional support. We also thank Lara Couto Baptista for her lab assistance, David Thieltges for his comments on the manuscript text, and the reviewers whose feedback helped improve the article.

## Author contributions

**Conceptualization:** Alessandro Messora, Monika Bright, Sabine Gollner.

**Data curation:** Alessandro Messora.

**Formal analysis:** Alessandro Messora.

**Funding acquisition:** Monika Bright, Sabine Gollner.

**Investigation:** Alessandro Messora, Fanny Sieler, Sabine Gollner.

**Methodology:** Alessandro Messora, Teresa Winter, Sabine Gollner.

**Project administration:** Monika Bright, Sabine Gollner.

**Resources:** Stephane Hourdez, Teresa Winter, Sabine Gollner.

**Supervision:** Stephane Hourdez, Sabine Gollner.

**Writing – original draft:** Alessandro Messora, Sabine Gollner.

**Writing – review & editing:** Alessandro Messora, Stephane Hourdez, Monika Bright, Teresa Winter, Fanny Sieler, Sabine Gollner.

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
