## [Decision Letter · Decision Letter 0]

10 Jul 2025

Dear Dr. Messora,

Thank you for submitting your manuscript to PLOS ONE. After careful consideration, we feel that it has merit but does not fully meet PLOS ONE’s publication criteria as it currently stands. Therefore, we invite you to submit a revised version of the manuscript that addresses the points raised during the review process.

I am pleased to inform you that your manuscript has undergone evaluation by two independent expert reviewers. Both reviewers concur that your manuscript is indeed highly interesting and commend its well-written nature.

The overall assessment is very positive, indicating that your work is a valuable contribution. The reviewers have identified only minor details that require attention and correction. These adjustments are primarily focused on refining certain aspects of the manuscript to enhance its clarity and precision.

Please address these specific points as outlined in the detailed reviewer comments. I anticipate that these minor revisions will further strengthen your submission

We look forward to receiving your revised manuscript.

Kind regards,

Clara F. Rodrigues

Academic Editor

PLOS ONE

Journal Requirements: 

Reviewers' comments:

Reviewer's Responses to Questions

**Comments to the Author**

1. Is the manuscript technically sound, and do the data support the conclusions?

Reviewer #1: Yes

Reviewer #2: Yes

2. Has the statistical analysis been performed appropriately and rigorously?

Reviewer #1: Yes

Reviewer #2: Yes

3. Have the authors made all data underlying the findings in their manuscript fully available?

Reviewer #1: Yes

Reviewer #2: Yes

4. Is the manuscript presented in an intelligible fashion and written in standard English?

Reviewer #1: Yes

Reviewer #2: Yes

Reviewer #1: Dear authors,

congratulations on a nice manuscript on an intresting (and as you say - difficult to examine) topic. I think your experimental work is nice and the results are both interesting and clearly presented. I am very happy to see some of the smaller animals focused on - even with all the extra difficulties a study of them brings. To that extent I was happy to see Figure 1 showing how utterly small these animals are - but maybe (since they are so tiny) it would be good if you could help the readers see them? (an arrow to a few specimens or something?) Personally I would also love to see photos (lab photos of specimens) of the different species you are discussing here - as this manuscript is of interest to more people than those already aquainted with the nice copepods. Could you do an extra figure with "the usual suspects" of your manuscript?

Even if I initially thought that doing a species identification after the experiment was not optimal I do understand why it had to be that way logistically, and even though in an ideal world I would still like to have the species separately tested (not as communities, even when that also gives interesting results), the nature of identification of these small lovely animals makes that impossible. I would still like to see some more discussion on what influence the more mixed community at the mussel/worm (diffuse) areas have at the timing of their collapse - could it be that some species were driving that timing (and the other species dying off earlier?) or do you think (all of the species represented in) the community responds similarly?

The only other thing I really would look at is the colour-use in your graphs (figs 3, 5 and 6 as well as app fig 2) - the combination of colours as well as maybe especially the green line makes seeing everything a bit difficult. Is a colour-change difficult?

Reviewer #2: This study investigates the thermal tolerance of copepod communities (Dirivultidae) inhabiting hydrothermal vent ecosystems along the East Pacific Rise, focusing on two habitat types distinguished by the intensity of hydrothermal fluid flow.

I have only a few comments to make, primarily aimed at clarifying specific points in the manuscript. Overall, the study is well-structured and presents valuable data.

Please clarify the definition of the term "focused flow habitat" in the Introduction, particularly in contrast to "diffuse flow habitat". While the term is mentioned in line 44, it remains unclear whether "focused flow" specifically refers to the high-temperature hydrothermal fluid (up to 120 °C) also mentioned.

Statistical methods: please specify what type of t-test was used, and whether normality, equal variances, were verified. Please also clarify the criteria for a "good fit" when reporting R² values. Values <0.70 are described as good, which should be justified or rephrased.

The manuscript currently repeats many statistical values across the main text, figure table and their legends (especially Fig. 3–6 & Tables 1–2). Please consider simplifying by presenting each value in one place and referencing it where necessary.

The authors could perhaps compare their results with hydrothermal vents from other regions in the discussion?

Abstract: rephrase "LT50 and LD50" to make it more accessible.

Line 36: and also volcanic arcs

Line 38: Microorganisms other than bacteria are also involved

Line 56: ‘.’ before ‘But’

Line 280: ‘Figure 2’ should likely be corrected to ‘Figure 3’.

Fig.2: In the legend, replace ‘In figure B’ with ‘In Tubeworm & Mussel habitat’ for clarity.

Fig.3 & 5: Indicate that the incubations were conducted at 200 bar.

Table 2: add ‘and anoxic’ conditions in the title

Tables 1 & 2: Add a description of the statistical tests used in the legends.

Suppl. Data: a column could be added to clearly identify each of the 118 incubations

**Do you want your identity to be public for this peer review?** For information about this choice, including consent withdrawal, please see our Privacy Policy

Reviewer #1: **Yes: ** Anne Helene S. Tandberg

Reviewer #2: No

---

## [Author Response · Author response to Decision Letter 1]

15 Sep 2025

Reviewer #1:

Dear authors,

congratulations on a nice manuscript on an interesting (and as you say - difficult to examine) topic. I think your experimental work is nice and the results are both interesting and clearly presented. I am very happy to see some of the smaller animals focused on - even with all the extra difficulties a study of them brings. To that extent I was happy to see Figure 1 showing how utterly small these animals are - but maybe (since they are so tiny) it would be good if you could help the readers see them? (an arrow to a few specimens or something?)

We thank Reviewer 1 for the positive comments on the manuscript’s topic and results. Dealing with very small animals also means that showing them in their habitat with pictures can be challenging. Since the copepods in Figure 1A are indeed very small reddish dots and easy to miss, we have added a few arrows inside the circle pointing to some of the largest groups of copepods. Figure 1B, instead, only shows a zoomed-out tubeworm & mussel clump and copepods are not visible.

Personally I would also love to see photos (lab photos of specimens) of the different species you are discussing here - as this manuscript is of interest to more people than those already acquainted with the nice copepods. Could you do an extra figure with "the usual suspects" of your manuscript?

We agree that including photos of the most common copepod species presented in our study would help readers who are not already acquainted with this topic. We do not have SEM or confocal images of the studied species but think that these show best how the Dirivultidae look like. We therefore added a representative picture of a Dirivultid copepod species in the supplementary materials.

Even if I initially thought that doing a species identification after the experiment was not optimal I do understand why it had to be that way logistically, and even though in an ideal world I would still like to have the species separately tested (not as communities, even when that also gives interesting results), the nature of identification of these small lovely animals makes that impossible.

We thank reviewer 1 for acknowledging the difficult logistics in deep sea research and dealing with very small animals. We appreciate the understanding, especially as this is the first study attempting this type of experiment on live deep-sea meiofauna.

I would still like to see some more discussion on what influence the more mixed community at the mussel/worm (diffuse) areas have at the timing of their collapse - could it be that some species were driving that timing (and the other species dying off earlier?) or do you think (all of the species represented in) the community responds similarly?

This is an interesting topic that unfortunately we cannot answer with our data and thus do not discuss in detail, but do acknowledge it now better in the discussion.

“However, it needs to be acknowledged that little is known on the experienced habitat conditions for copepods in the tubeworm & mussel habitat and how the ecological niches of each species impact their exposure to elevated temperature and low oxygen concentrations. Only more detailed abiotic measurements coupled with species-specific survival curves would show if the thermal tolerances of each species represent a shared response akin to that of the broader community or if instead the community results are a weighted average of each species’ thermal tolerance.”

The only other thing I really would look at is the colour-use in your graphs (figs 3, 5 and 6 as well as app fig 2) - the combination of colours as well as maybe especially the green line makes seeing everything a bit difficult. Is a colour-change difficult?

We agree that the use of colour in the graphs mentioned by reviewer 1 made interpretation a little difficult. Now the bright primary colours of the survival curves have been substituted with more solid colours that should increase readability. In addition, we avoid using green considering red-green color blindness.

Reviewer #2:

This study investigates the thermal tolerance of copepod communities (Dirivultidae) inhabiting hydrothermal vent ecosystems along the East Pacific Rise, focusing on two habitat types distinguished by the intensity of hydrothermal fluid flow.

I have only a few comments to make, primarily aimed at clarifying specific points in the manuscript. Overall, the study is well-structured and presents valuable data.

Please clarify the definition of the term "focused flow habitat" in the Introduction, particularly in contrast to "diffuse flow habitat". While the term is mentioned in line 44, it remains unclear whether "focused flow" specifically refers to the high-temperature hydrothermal fluid (up to 120 °C) also mentioned.

We thank reviewer 2 for the positive comments on the manuscript’s presentation and results. Focused flow hydrothermal fluid is the superheated solution exiting from chimney spires that in our study area reaches roughly 120°C as it exits the vent chimney (while still inside the chimneys of black smokers in the study area, they can however exceed ~350°C). The surfaces in direct contact with these fluids are not inhabited by animals. Focused flow habitats, instead, refer to the area/habitat surrounding the chimney’s outer surface where turbulent mixing occurs between the warm fluid and the surrounding 2°C seawater. By definition, the temperature of this habitat will constantly fluctuate between these two extremes somewhat proportionally to how far away you are situated from the vent opening. Exactly how frequently animals in this habitat are exposed to very hot fluids is difficult to say, but from other studies cited in the manuscript we can conclude that the outer surface of this habitat tends to average 30°C, rarely surpasses 40°C and temperatures approaching 120°C are very rarely experienced .

The sentence in the introduction between lines 42-47 has been edited to clarify the definition of a focused flow habitat: “Along the East Pacific Rise (EPR), the tubicolous Pompeii worm Alvinella pompejana [7, 8] thrives in focused flow habitats, where the hot hydrothermal fluids (up to 120°C) at the smoker’s surface undergo a substantial temperature decrease to 30°C, helped by the colony’s physical and chemical gradients, and mix with the surrounding 2°C seawater [9, 10]”.

Statistical methods: please specify what type of t-test was used, and whether normality, equal variances, were verified. Please also clarify the criteria for a "good fit" when reporting R² values. Values <0.70 are described as good, which should be justified or rephrased.

We thank reviewer 2 for the comment and we rechecked all data and tests. Testing for normality (Shapiro-Wilk test) and homoscedasticity (Breusch-Pagan test) in our dataset showed that the assumption of normality was maintained when comparing intercepts and slopes of TDT curves. Comparisons between LD50 survival curves did not reveal normality, and wild bootstrapping method with 2000 iterations, which does not necessitate for a normal distribution of the data, was chosen to compare survival curves. This newly improved statistical analysis did not change the results. The methods of the revised manuscript have been updated to include the packages used for data analysis. In addition, whenever t-tests were implemented, we made sure to specify that they were Welch’s t-tests and showed that the data was normally distributed. Finally, we rephrased terminology as suggested by the reviewer with regards to the criteria for good R2 fitting as acceptable between 0.6 and 0.7 and good above 0.7.

The paper now reads as such:

“Curve fitting was considered of good quality for R2 values higher than 0.7 and of acceptable quality for values between 0.6 and 0.7. Model adequacy was assessed through residual analysis, including tests for normality (Shapiro-Wilk test) and homoscedasticity (Breusch-Pagan test). After confirming normality assumptions (normality: p > 0.05; homoscedasticity: p > 0.05), Welch's t-tests were performed between the TDT slopes and intercepts of the two vent habitats to determine the statistical significance of the observed differences. Comparisons between LD50 values of oxic/anoxic and pressurized/unpressurized 2 h survival curves for each habitat contained residuals that violated normality or homoscedasticity assumptions. In these cases, wild bootstrap resampling was employed as a robust alternative [61]. Wild bootstrap preserves the fitted model structure while resampling residuals, ensuring a realistic estimate of LD50 in all resamples. P-values were calculated from 2000 iterations, providing distribution-free inference robust to non-normal error structures commonly observed in thermal tolerance experiments and allowing to determine the statistical significance of the effect of anoxia and pressure on copepod survival.”

The manuscript currently repeats many statistical values across the main text, figure table and their legends (especially Fig. 3–6 & Tables 1–2). Please consider simplifying by presenting each value in one place and referencing it where necessary.

We thank the reviewer for this suggestion. Redundant statistical values have been removed, especially from figure captions, instead opting to keep them confined to Tables 1 and 2 and either mentioning them more sparingly or referring directly to the tables.

The authors could perhaps compare their results with hydrothermal vents from other regions in the discussion?

We thank the reviewer for the suggestion. However, when broadening our scope and especially in the context of deep-sea physiology, very little to no data is available. To our knowledge, there are no other studies that empirically tackle the thermal physiology of vent meiofauna beyond ours and those we cite, such as Sell, 2000 and Hourdez et al., 2000. Several physiological studies have been performed on vent macro and megafauna and were included in our discussion, as the shared component of habitat and physico-chemical conditions between size classes contributes to the knowledge. However, due to the different nature of experiments carried out, a more direct comparison of results would risk misinterpretation, which we prefer to avoid.

Abstract: rephrase "LT50 and LD50" to make it more accessible.

We thank the reviewer and agree that rephrasing this part of the abstract improves accessibility. The terms LT50 and LD50 have been substituted with median lethal time and median lethal temperature respectively.

Line 36: and also volcanic arcs

The occurrence of vents along volcanic arcs has now been added. The orginal citation was kept as it refers also to arcs.

Line 38: Microorganisms other than bacteria are also involved

The term bacteria has been replaced with microbes. The reference cited remains the same as it also refers to microorganisms more broadly.

Line 56: ‘.’ before ‘But’

The sentence has now been rephrased to improve readability. We added a “dot” and started a new sentence. “In addition, vents are subject to a long-term cycle….”

Line 280: ‘Figure 2’ should likely be corrected to ‘Figure 3’.

Thank you for spotting this typing error, and it has now been corrected.

Fig.2: In the legend, replace ‘In figure B’ with ‘In Tubeworm & Mussel habitat’ for clarity.

The sentence has now been rephrased as suggested to improve readability.

Fig.3 & 5: Indicate that the incubations were conducted at 200 bar.

These sentences have now been rephrased to add this information.

Table 2: add ‘and anoxic’ conditions in the title

This sentence has now been rephrased to add this missing information.

Tables 1 & 2: Add a description of the statistical tests used in the legends.

The legends of both tables have now been edited to add this missing information.

Suppl. Data: a column could be added to clearly identify each of the 118 incubations

An extra column has now been added to better visualize which samples were incubated together as replicates.

---

## [Editor Report · Decision Letter 1]

22 Sep 2025

Limits of life: thermal tolerance of deep-sea hydrothermal vent copepods and implications for community succession

PONE-D-25-24603R1

Dear Dr. Messora,

We’re pleased to inform you that your manuscript has been judged scientifically suitable for publication and will be formally accepted for publication once it meets all outstanding technical requirements.

Kind regards,

Clara F. Rodrigues

Academic Editor

PLOS ONE

Additional Editor Comments (optional):

Thank you for addressing all the questions raised during the review process
---

## [Editor Report · Acceptance letter]

PONE-D-25-24603R1

PLOS ONE

Dear Dr. Messora,

I'm pleased to inform you that your manuscript has been deemed suitable for publication in PLOS ONE. Congratulations! Your manuscript is now being handed over to our production team.

Kind regards,

on behalf of

Dr. Clara F. Rodrigues

Academic Editor

PLOS ONE